# Computer-Aided Detection of Fiducial Points in Seismocardiography through Dynamic Time Warping

**DOI:** 10.3390/bios12060374

**Published:** 2022-05-30

**Authors:** Chien-Hung Chen, Wen-Yen Lin, Ming-Yih Lee

**Affiliations:** 1Graduate Institute of Biomedical Engineering, Chang Gung University, Taoyuan 33302, Taiwan; d0528005@cgu.edu.tw (C.-H.C.); leemiy@mail.cgu.edu.tw (M.-Y.L.); 2Center for Biomedical Engineering, Department of Electrical Engineering, Chang Gung University, Taoyuan 33302, Taiwan; 3Division of Cardiology, Department of Internal Medicine, Chang Gung Memorial Hospital, Taoyuan 33302, Taiwan

**Keywords:** cardiac time interval, dynamic time warping, fiducial point detection, heart failure, seismocardiography

## Abstract

Accelerometer-based devices have been employed in seismocardiography fiducial point detection with the aid of quasi-synchronous alignment between echocardiography images and seismocardiogram signals. However, signal misalignments have been observed, due to the heartbeat cycle length variation. This paper not only analyzes the misalignments and detection errors but also proposes to mitigate the issues by introducing reference signals and adynamic time warping (DTW) algorithm. Two diagnostic parameters, the ratio of pre-ejection period to left ventricular ejection time (PEP/LVET) and the Tei index, were examined with two statistical verification approaches: (1) the coefficient of determination (R^2^) of the parameters versus the left ventricular ejection fraction (LVEF) assessments, and (2) the receiver operating characteristic (ROC) classification to distinguish the heart failure patients with reduced ejection fraction (HFrEF). Favorable R^2^ values were obtained, R^2^ = 0.768 for PEP/LVET versus LVEF and R^2^ = 0.86 for Tei index versus LVEF. The areas under the ROC curve indicate the parameters that are good predictors to identify HFrEF patients, with an accuracy of more than 92%. The proof-of-concept experiments exhibited the effectiveness of the DTW-based quasi-synchronous alignment in seismocardiography fiducial point detection. The proposed approach may enable the standardization of the fiducial point detection and the signal template generation. Meanwhile, the program-generated annotation data may serve as the labeled training set for the supervised machine learning.

## 1. Introduction

The emergence of wearable solutions for health monitoring provides opportunities for remote medical surveillance. Biosensing and cloud computing technologies enable physiological parameters to be tracked unobtrusively, accurately, and in real time in everyday life. In most cases, relatively simple, reproducible, and reliable generic physiological parameters are monitored. Cardiac time intervals (CTIs), durations between specific cardiac events, are closely related to cardiac physiology and function. CTIs play a pivotal role in the diagnostic and prognostic assessments of patients with hemodynamic and valve dysfunction, especially with regard to risk stratification [1,2].

In clinical practice, CTIs associated with valve opening and closure, including the pre-ejection period (PEP), left ventricular ejection time (LVET), isovolumic contraction time (IVCT), and isovolumic relaxation time (IVRT), are employed in the calculation of the myocardial health index. These CTIs describe the time periods between the specific cardiac events of mitral valve opening (MO), mitral valve closure (MC), aortic valve opening (AO), aortic valve closure (AC), and Q wave in electrocardiography (ECG). Four frequently used CTIs are defined by the timing differences between cardiac events as follows:PEP = AO *−* Q,(1)
LVET = AO *−* AC,(2)
IVCT = AO *−* MC,(3)
IVRT = MO *−* AC.(4)

Although some of the CTIs are modulated by heart rate, respiration, or even the individual’s posture and position [3,4], the cardiac indexes derived from CTIs are considered reliable parameters for predicting and assessing myocardial contractility [2,3,5]. Two frequently used cardiac indexes are the PEP/LVET ratio (also known as the contractility coefficient) [6,7] and the Tei index (also known as the myocardial performance index) [5]. They are defined as follows:(5)Contractility Coefficient = PEP/LVET,
(6)Tei index =(IVCT + IVRT)/LVET.

Both the PEP/LVET ratio and the Tei index have been clinically confirmed to be heart rate-independent and negatively correlated with the left ventricular ejection fraction (LVEF) on a beat-to-beat basis [2,3,5]. Carvalho observed that individuals with normal cardiac function (higher LVEF) exhibited a short PEP and a long ejection time. By contrast, patients (lower LVEF) with reduced stroke volume (SV) and cardiac output (CO) had a longer PEP and shorter LVET [8]. Thus, CTIs and the two cardiac indexes mentioned above can supplement the LVEF and serve as proxies in the assessment of myocardial contractility [9,10,11].

Echocardiography is considered the gold standard for obtaining CTIs [8]. CTIs are typically acquired using ultrasound modalities such as color Doppler flow imaging, tissue Doppler imaging, or M-mode echocardiography [8,12,13,14]. From echocardiography images, physicians can accurately determine the timings of the start, peak, and end of specific hemodynamic or cardio-mechanical events (e.g., blood ejection, myocardial motion, and valve opening or closure) in various phases of the cardiac cycle. However, acquiring echocardiograms is time consuming and requires the expertise of a well-trained sonographer. Therefore, for long-term, home-based cardiac monitoring, the use of comparatively simple, noninvasive wearable devices to conduct hemodynamic assessment, such as through impedance cardiography, phonocardiography, and seismocardiography (SCG), is preferred [7,10,12].

SCG, a noninvasive approach for the diagnosis of cardiac conditions, is capable of evaluating CTIs through chest wall vibration analysis. Temporal information of cardiac events can be obtained by identifying the specific fiducial points in the SCG signals corresponding to the events. Single-channel and multichannel SCG monitoring systems and SCG–echocardiography hybrid apparatuses have been proposed [7,15,16]. In 1994, Crow examined trimodal screenshots of simultaneous ECG and SCG signals and echocardiography images to investigate the relationship between SCG and echocardiographic images regarding cardiac events [16]. In the trimodal measurements, the SCG signal was routed to the auxiliary input port of the ultrasound machine and was presented synchronously on top of the echocardiogram together with the ECG signal. This enabled the SCG fiducial point detection and the CTI analysis with the same heartbeat cycle on the same ultrasound image.

Home-based cardiac monitoring requires special analytical methods because no sonographers or ultrasound instruments are available. Heterogeneous modality cooperation may serve as the alternative; that is, conducting the diagnostic assessment using ECG and echocardiogram (also assisting SCG fiducial point identification) while home monitoring using ECG and SCG. Lin et al. introduced a quasi-synchronization method for SCG fiducial point detection from the SCG and ECG data collected through the inertial-sensor-based multichannel SCG systems and from several echocardiogram images acquired at different time [7,9]. The ECG signal from SCG measurement and the ECG signal from echocardiogram image were aligned by the manipulation of uniform stretching (or squeezing) and shifting in time axis to align the ECG R peaks to each other. This approach is referred to as the quasi-synchronous alignment (or “conventional” quasi-synchronous alignment) in this paper and is illustrated in Figure 1.

The development of SCG has been limited by artifact effects, the ambiguity in specific event waveforms and the lack of detection procedures; no standards for fiducial point detection in SCG signals have been established [17]. In this study, the problems of quasi-synchronous alignment were revisited, and an improved SCG fiducial point detection protocol was devised by introducing a reference SCG signal and the dynamic time warping (DTW) algorithm. The new alignment method is referred to as the “DTW-based” quasi-synchronous alignment, so as to distinguish it from the existing quasi-synchronous alignment method (or the “conventional” quasi-synchronous alignment). Compared to the conventional quasi-synchronous alignment, DTW-based quasi-synchronous alignment eliminates the fiducial point detection error due to the stretching (or squeezing) manipulation. Compared to the envelope-based detection methods [14,18,19,20], DTW-based quasi-synchronous alignment does not limit to any specific fiducial points, as long as the corresponding cardiac events could be identified in an clinical imaging modality (not limited to echocardiogram, that was used in this paper) or cardiac signal that is simultaneously measured with ECG. This method has the potential to untie the knot that limits the development of SCG.

As the deterministic approach in pattern recognition technology, by which machine learning was enabled in many applications [21,22], DTW is known for its capability to align the morphological patterns of two given time series (signals), its flexibility in handling signals of varying length, and its feasibility of implementation through computer programs [23]. DTW has been applied to speech recognition, time series clustering, and protein sequence alignment [24,25,26]. Herein, DTW was employed to mitigate the waveform variations in SCG signal analysis. In Azad’s study, the DTW distance, instead of the Euclidean distance, significantly reduced the morphological variability in clustered SCG signals corresponding to groups of participants in various breathing stages [27]. Based on DTW alignment, an investigation on subject-oriented template generation succeeded to establish the procedures for signal clustering (by using the k-means algorithm) and averaging according to SCG morphological features for the first time [28].

The remainder of this paper is organized as follows. In Section 2, the data acquisition setup, data preprocessing procedure, and the reason for introducing the reference SCG segment are declared. The experimental results are provided in Section 3. The discussions and conclusions are presented in Section 4.

## 2. Materials and Methods

The study protocol of this work was approved by the Institutional Review Board of Chang Gung Memorial Hospital, Taoyuan, Taiwan (approval number 202100744B0A3). The physiological data (112 echocardiography images and SCG signal clips of 3876 heartbeats) of 56 individuals (30 men, 26 women) collected in the previous study by Lin [9] on a multichannel SCG spectrum system were employed in this study. The data for each individual comprised the diagnostic history, LVEF assessment, echocardiography images, ECG signals, and a set of four-channel SCG signals. In this study, only the first channel SCG data was analyzed, and the signal was recorded from the location of the fifth-left intercostal space in the midclavicular line of the mitral valve with the participant in a supine position. The echocardiogram images captured the parasternal long axis (PLAX) in the M-mode and the Doppler flow images of the mitral and aortic valves, providing the timings of valve opening and closure. Participants were instructed not to exercise before the tests. The raw SCG and ECG data underwent filtering, detrending, wavelet noise reduction, cardiac cycle identification, ECG wave annotation, and cycle segmentation [28].

The following sections introduce concepts and problems of the conventional quasi- synchronous alignment, as well as concepts and advantages of DTW-based quasi-synchronous alignment. Furthermore, a programming flowchart is presented and performed to validate the annotation procedures and the effectiveness of SCG fiducial point detection.

### 2.1. Conventional Quasi-Synchronous Alignment for Echocardiogram Image and SCG Signal

Although beat-by-beat synchronicity can be achieved in trimodal (echocardiography/ECG/SCG) measurements, the ultrasound probe may interfere with SCG sensors on the chest during such simultaneous measurements. Through simple shifting and rescaling (stretching or squeezing) manipulations on the record signals, the “asynchronously” measured echocardiogram images and SCG signals could be analyzed in the same graph to establish quasi-synchronization [9]. This method, which is as effective as trimodal measurement, caters to the increasing demand for the home-based monitoring for patients with cardiovascular disease and heart failure.

An example of the conventional quasi-synchronous alignment is displayed in Figure 1. The figure consists of an M-mode image of aortic valve motion (with synchronous ECG signal shown overlapped in the lower part of the image) (Figure 1a), an ECG (Figure 1b) and SCG (Figure 1c) signal pair measured at the same time. To align the echocardiogram image and the SCG signal quasi-synchronously, R peaks from the two ECG signals were employed as the beacon targets. The ECG signal in Figure 1b was rescaled and shifted to align the R peaks to the ECG R peaks of the echocardiogram image (in Figure 1a) as indicated in the orange boxes. The vertical blue lines in Figure 1 link the timing information between the echocardiogram image and SCG signals. As the events visually identified in the echocardiogram image, the timing positions of the fiducial points in SCG signal could be easily obtained. Therefore, the fiducial points of the SCG signals could be detected. This technique had also been applied to color Doppler echocardiogram and tissue Doppler echocardiogram images to identify six new fiducial points [9].

### 2.2. Misalignmenst and Detection Errors in SCG Fiducial Point Detection under Conventional Quasi-Synchronous Alignment

Although the conventional quasi-synchronous alignment is a rapid and intuitive approach in which ECG signals from echocardiogram and SCG measurements are used as the intermedium, the rescaling manipulation can be problematic. Waveform distortion due to rescaling may result in unexpected target shifting as the signals were subjected to the conventional quasi-synchronous alignment. For easier graphic illustration and better clarification, two ECG and SCG signal pairs with different heartbeat lengths were used as examples in Figure 2, Figure 3 and Figure 4, whereas the mostly concerned targets in this study are echocardiogram image and SCG signal. The issue exists in spite of the target change as long as the end-to-end alignment was achieved by the rescaling manipulation.

Figure 2 demonstrates the conventional quasi-synchronous alignment involving the stretching manipulation of short-period ECG and SCG signals such that they were matched end to end with long-period signals. Figure 3 displays the same alignment approach but performed by squeezing the long-period signals such that they were matched end to end with the short-period signals. The principal problem of the conventional quasi-synchronous alignment is applying a unique rescaling ratio (stretching or squeezing scale ratio) to the entire waveform. The unique rescaling ratio manipulation is suspicious as all the fiducial points (MO, MC, AO, and AC) of the short-period signal (blue line) in Figure 2 and Figure 3 are very close to the corresponding fiducial points in the long-period signal (red line). Over both the short and long period heartbeats, the heart pulsed at a comparable pace in the beginning (before 600 ms). Overall, length and morphology differences were noted in the heartbeat signals; this is attributed to the deviation in the signal length caused by the unequal cardiac paces in the latter portion of the cycle (after 600 ms). The widths of the yellow bars in Figure 2b and b represent the variations in SCG fiducial point timings between the short- and long-period signals before the rescaling manipulation, which are ≤10 ms.

Three types of lines (red solid, blue solid, and black dashed) annotated with four types of icons (squares, triangles inverted triangles, and circles) are shown in Figure 2b and Figure 3b. The icons and lines presented in blue and red correspond to the pre-detected fiducial points and SCG signals for the short- and long-period heartbeats, respectively. Figure 2b illustrates the distorted signal (black dashed curve) and timing push off in fiducial points (black icons) ascribable to the uniform stretching process (indicated by the sign of blue rightwards arrow). The amounts of detection error for the four fiducial points are indicated by the lengths of the white arrows. Similar detection errors of the fiducial points are observable in Figure 3b, whereas the long-period signal is squeezed to match the short-period signal. The white arrows are with the lengths (of several tens milliseconds) considerably longer compared to the widths of the yellow bars, and with a trend that the longer arrows appear closer to the right end. The trend reveals that the conventional quasi-synchronous alignment led to different detection errors for different fiducial points. This is owing to the signal morphological distortion as the left end is fixed while applying a unique rescaling ration to the entire signal to make the right end match that of another signal with a different length. Likewise, if the SCG fiducial points detected from the SCG-echocardiogram alignment adopt the conventional quasi-synchronous approach, the same signal distortion and detection errors will occur. That will degrade the CTI assessment and the diagnostic applications of SCG.

### 2.3. Introduction of Reference Signal in Quasi-Synchronous Alignment

Figure 4 illustrates two ECG and SCG waveforms with the same heart cycle period spontaneously aligning to each other, not only in the former half section but also in the latter half section. This phenomenon (the comparable heart pace and self-alignment in morphology over the entire heart cycle) was leveraged in the proposal of this study (a new quasi-synchronous alignment method) to avoid the signal distortion and to mitigate the misalignments in SCG fiducial point detection. For this reason, a reference SCG signal was introduced as the intermediary to accommodate the fiducial points detected from the alignment to echocardiogram images and also to project the fiducial points to other nonreference SCG signals.

**Figure 4 biosensors-12-00374-f004:**
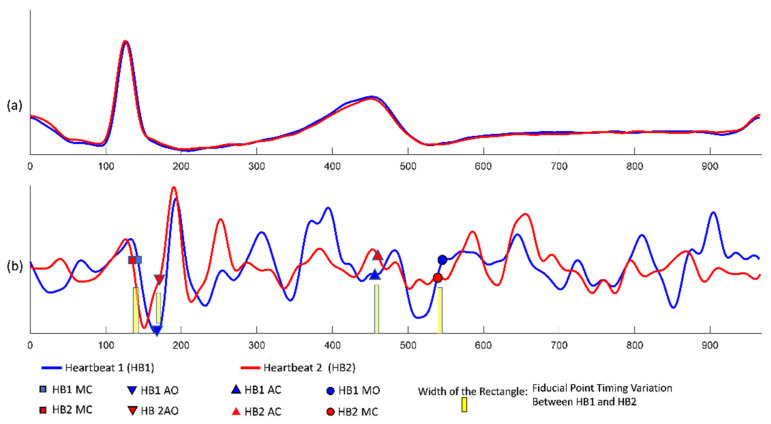
Spontaneous alignment of two ECG and SCG signals with the same heart cycle period: (**a**) ECG signals; (**b**) SCG signals with the detected fiducial points.

To avoid the waveform distortion during the signal alignment, a reference signal pair of ECG and SCG were screened out from the SCG measurements which had the heart cycle period closest to that of the ECG signal in the echocardiogram image. In Figure 5, a reference signal pair example is illustrated, both the selected RR intervals of the ECG signals in echocardiogram (Figure 5a) and the RR intervals in the reference signal pair (Figure 5b) are the same (966 milliseconds). As the RR interval of the reference ECG signal performing end-to-end alignment with the echocardiogram ECG, no rescaling manipulation was needed. In addition, the entire ECG and SCG signals were considered synchronous to the selected cycle in the echocardiogram on the premise of spontaneous alignment discussed in Figure 4. The aortic valve closing event (annotated by “AC” in Figure 5) identified in the echocardiogram, was mapped to the same temporal place in the reference ECG and SCG signals. The reference SCG signal with fiducial points mapped from the echocardiogram was then used as the intermediary template signal for fiducial point projection to other nonreference measurements. The projection process based on the DTW algorithm, which takes signal morphological similarity into consideration. The revised quasi-synchronous alignment method hereinafter referred to as DTW-based quasi-synchronous alignment.

### 2.4. DTW

DTW, a dynamic programming base algorithm, has been applied to the speech recognition analysis with the identical voice content but morphologically varied at different speech paces [29]. The flexibility of DTW algorithm in aligning two sequences (not necessarily equal in data length) extends its application from speech recognition to biometrics (e.g., fingerprints), handwriting and many other technological fields [30].

Given two morphologically similar signal sequences, *X = (x*_1_, *x*_2_,..., *x_M_)* of length *M (M* ∈ *ℕ)* and *Y = (y*_1_, *y*_2_,*…*, *y_N_)* of length *N (N* ∈ *ℕ)*, the goal of the DTW algorithm is to align *X* and *Y* to optimize distance cost function requirements as well as to conform to the warping path constraints of (1) monotonicity, (2) continuity, (3) boundary, (4) warping window, and (5) slope constraints [31,32]. A cost matrix with size of *M × N* was constructed to present all possible path points of the dynamic warping. Given a path *W = (w*_1_, *w*_2_,..., *w_K_)* of length *K*, *max(M*,*N)* ≤ *K* ≤ *M + N − 1*. Any *w_i_* in *W* contains two index elements, *w_i_ = (a_i_*, *b_i_)*, with the first and second indexes corresponding to the *a_i_*th and *b_i_*th elements from *X* and from *Y*, respectively. Figure 6a illustrates the matrix grids of possible path steps and the aligned warping path for two sequences. Figure 6b shows the point-to-point mapping of the aligned signal sequences. The iterative dynamic planning process drove the warping path *W* gradually close to the optimized path as the distance cost function was tuning to the minimum scenario. Typically, the DTW distance cost function is defined as follows:(7)$$\begin{array}{c} Distance\;Cost\;Function\left(W\right)\\ =DTW\left(X,Y\right)\\ =\frac{1}{K}\sum_{i=1}^{K}D\left(w_{i}\right)\\ =\frac{1}{K}\sum_{i=1}^{K}\sqrt{{\left(x_{a_{i}}-y_{b_{i}}\right)}^{2}} \end{array}$$

Usually more than one path in the matrix grid can satisfy the requirements of the five constraints, but only one path minimizes the distance cost. A few different types of distance cost function variants have been developed for different application [33,34].

Some DTW applications concern only the Euclidean distance of the aligned points in the signal sequences (as in Equation (7)), whereas in the present study, the morphological similarity was most interested in and critical to fiducial point projection. A hybrid form of the cost function (termed the new cost function) was proposed (as in Equation (8)) because it not only considered the difference in signal values (item led by *β*) but also evaluates the neighborhood shifting level (item led by *α*), the difference in signal slope (item led by *γ*), and the difference in signal concavity (item led by *η*). The weighting factors *(α*, *β*, *γ*, and *η)* were tunable and might change across individuals.
(8)$$\begin{array}{c} New\;Cost\;Function\left(W\right)\\ =\frac{1}{K}\sum_{i=1}^{K}\left(\alpha \sqrt{{\left(a_{i}-b_{i}\right)}^{2}}+\beta \sqrt{{\left(x_{a_{i}}-y_{b_{i}}\right)}^{2}}+\gamma \sqrt{{\left(\frac{dx_{a_{i}}}{dt}-\frac{dy_{b_{i}}}{dt}\right)}^{2}\gamma +}\eta \sqrt{{\left(\frac{d^{2}x_{a_{i}}}{dt^{2}}-\frac{d^{2}y_{b_{i}}}{dt^{2}}\right)}^{2}}\right) \end{array}$$

### 2.5. Fiducial Point Projection with DTW-Based Quasi-Synchronous Alignment

The DTW-base quasi-synchronous alignment introduces two modifications to the conventional quasi-synchronous alignment: (1) the intermediary reference signal pair, and (2) DTW-based point-to-point alignment of two signal sequences. The examples and comparison of SCG fiducial point detections in the nonreference SCG signal using the conventional and DTW-based quasi-synchronous alignment methods are demonstrated in Figure 7. Figure 7a–c illustrate the conventional approach, whereas Figure 7d–g display the DTW-based approach. The echocardiogram shown in Figure 7a is a color Doppler flow measurement at the aortic valve. In Figure 7a, the 966-ms period between R1 and R2 is selected as the synchronization target. The signals in Figure 7b,d represent the same nonreference ECG signal, whereas Figure 7b is rescaled (squeezed to fit signal length of 1003 ms to 966 ms) and shifted such that it can be visually aligned to the ECG R1-R2 section in Figure 7a. The signals in Figure 7c,f represent the same SCG signal (other than the reference SCG signal). Figure 7c has been rescaled and synchronized to Figure 7b.

#### 2.5.1. Common Procedures under Both Alignment Methods 

An RR interval in the echocardiogram image was selected and the ends were annotated as R1 and R2. A visually recognized AC event was marked by a blue line and labeled as “AC” in the echocardiogram (Figure 7a). The present RR interval (966 ms) was obtained by pixel counting from the timing ticks in the echocardiogram.

#### 2.5.2. Conventional Quasi-Synchronous Alignment

The SCG and ECG signals were shifted and rescaled to align R peaks to R1 and R2 in the ECG of echocardiogram (Figure 7a–c). The fiducial point was obtained by extending the blue line from Figure 7a–c.

#### 2.5.3. DTW-Based Quasi-Synchronous Alignment

A reference signal pair of ECG and SCG was sorted out with the RR interval closest to the R1–R2 period. In this example, the reference signal pair had a heartbeat cycle of 966 ms. Shifting, but not rescaling, was required to align the reference signals until the ECG R peaks matched R1 and R2, as shown in Figure 7e,g. Referring to the condition in Figure 4, the ECG signals in Figure 7a,e, as well as the echocardiogram and SCG signal in Figure 7g, are postulated to align to each other (within the RR interval) spontaneously. The fiducial point AC was detected in the reference SCG signal by extending the blue line from Figure 7a–g. The fiducial point AC was projected from the reference SCG signal (Figure 7g) to nonreference SCG signal (Figure 7f) with the aid of a DTW-based software program, as shown in Figure 7f–i.

As mentioned earlier, a fiducial point detection error resulted from the signal rescaling manipulation in the conventional quasi-synchronous alignment, as indicated by the width of the yellow rectangle in Figure 7h. Figure 7i displays the results of DTW point-to-point alignment with the new cost function between the reference SCG signal (red) and the target nonreference signal (blue).

### 2.6. Validation of DTW-Based Fiducial Point Detection Approach

An SCG fiducial point detection software tool was developed, using MATLAB R2020a, following the programming flowchart (in Figure 8) which depicts the SCG fiducial point detection procedures with DTW-based quasi-synchronous alignment. Two cardiac diagnostic parameters (PEP/LVET ratio and Tei index) were extracted from a dataset of 56 individuals to assess the clinical application of the proposed SCG fiducial point detection method and to validate the effectiveness of the software. In addition, Figure 8 also reveals that the correlation of averaged diagnostic parameters versus LVEF and ROC classification analysis were conducted at the end of the proof-of-concept experiment.

#### 2.6.1. Clinical Data Acquisition and Preprocessing

At least two echocardiogram images were required to detect all the interested cardiac events (MO, MC, AO, and AC) because an ultrasonic probe can only examine one cardiac valve at a time. A set of simultaneously measured ECG and SCG signal pairs were clipped from the continuous SCG measurement. As shown in Figure 7b,c, the ECG and SCG signal clips were sectioned according to ECG T waves, from T0 to T2, to include a complete heartbeat section of the centered RR interval and with extra intervals before and after. Prior to the fiducial point detection, signal clips went through the data preprocessing including signal detrending, band-pass filtering, and wavelet denoising.

#### 2.6.2. Echocardiogram–Reference Signal Alignment

An RR interval in the echocardiogram was selected for the interested cardiac event identification as shown in the section confined by R1 and R2 lines in Figure 7a. To minimize the fiducial point detection error, an ECG/SCG signal pair with RR interval period the same as or close to the duration between R1 and R2 lines were screened out from the signal clips as the reference signal pair. The reference signal pair were aligned to the R1–R2 section in the echocardiogram only through shifting, as presented in Figure 5 or in Figure 7e,g.

#### 2.6.3. Identification and Mapping of Cardiac Events in Echocardiogram to Reference Signals 

After the alignment of echocardiogram images and reference signals, the valve opening and closure events were identified and annotated in the images, as indicated by the blue line in Figure 5a or Figure 7a. By extending the lines to reference SCG signals, the intersections were annotated as the detected SCG fiducial points in the reference signal.

#### 2.6.4. Projection of Fiducial Points from Reference to Nonreference Signals

The reference SCG signals with fiducial point annotations were used to project the fiducial points to nonreference signals through DTW alignment, under the proposed new cost function optimization. Figure 6 illustrates the results of the point-to-point mapping of the alignment and the aligned path with minimized cost function in the cost matrix.

#### 2.6.5. PEP/LVET Ratio and Tei Index Calculation

To validate the clinical practicability and effectiveness of the DTW-based fiducial point detection approach, the cardiac parameters (PEP/LVET ratio and Tei index) for each nonreference SCG signal clip were calculated according to Equations (5) and (6) from the projected fiducial points.

#### 2.6.6. Index Statistics Calculation

To assess the cardiac health of an individual, the averaged PEP/LVET ratio and Tei index were utilized instead of using the indexes from a single signal clip.

#### 2.6.7. Correlation of the Averaged Indexes to the LVEF

The collection of the averaged indexes (PEP/LVET ratio and Tei index) for 56 individuals were correlated to the individual’s clinical LVEF assessment (in Figure 9). The coefficient of determination (R^2^) was employed as the indicator of the correlation between the averaged indexes and the clinical LVEF assessment.

#### 2.6.8. ROC Curve Analysis

Four predictive models in ROC curve analysis (Figure 10) were used to distinguish the patients of heart failure with reduced ejection fraction (HFrEF), using the predictors of (1) clinical LVEF assessment, (2) PEP/LVEF ratio, (3) Tei index, and (4) the mean PEP/LVEF ratio and Tei index, respectively. The values of area under the ROC curve (AUC) were determined, to evaluate the predictability of the models as well as the effectiveness of these diagnostic parameters derived from the DTW-base quasi-synchronous alignment.

## 3. Results

With IRB approval, the SCG–echocardiogram data of the 56 individuals were employed for the proof-of-concept experiments and the effectiveness evaluation of DTW-based fiducial point detection approach. The experimental data were reused from the previous research conducted by Lin et al. on a multichannel SCG system [9].

Table 1 lists the subjects’ demographic information and SCG-derived cardiac parameters, including patient ID, sex, age, clinical LVEF assessment, hospital diagnosed cardiovascular diseases, the number of examined SCG clips and the statistics of PEP/LVET ration and Tei index derived by the DTW-based SCG fiducial point detection method. The means ± standard deviations of age and LVEF are 52.1 ± 22.3 years and 50.8% ± 16.3%, respectively. The mean and standard deviation of PEP/LVET ratio and Tei index were calculated from the number of examined SCG clips. The number of data clips used for fiducial point detection ranged from 34 to 106, with a mean ± standard deviation of 69.2 ± 16.2.

The applicability of SCG fiducial point detection using DTW-based alignment was validated through two experiments: (1) the linear correlation between SCG-derived indexes (PEP/LVET ratio and Tei index) and clinical LVEF assessment and (2) the analysis of ROC classification to distinguish patients with HFrEF or else.

### 3.1. Linear Correlation Models

The clinical LVEF assessment has been reported to be negatively proportional to the PEP/LVET ratio and Tei index for the patients with cardiac symptoms of varying severity [3,5,7,35,36]. To validate the effectiveness of the SCG fiducial points and the associated CTIs derived using the DTW-based quasi-synchronous alignment method, three general linear models (GLMs) were generated for the correlation analysis to prove the negative proportionality. The trendlines of the 56-subject clinical LVEF assessments were synthesized under three general linear regression models by using the univariate predictors of: (1) the PEP/LVET ratio, (2) Tei index, and (3) the mean PEP/LVET ratio and Tei index (shown in Figure 9). The three general linear regression models are formulated as Equations (9), (10), and (11). Favorable coefficients of determination (R^2^) for the three univariate linear models were obtained in this 56-subject experiment, with R^2^ = 0.768, 0.86, and 0.894 for (1), (2) and (3), respectively. The negative proportionalities in the graphs of Figure 9 are obvious, whereas the standard deviation (indicated by the blue error bars) and 95% confidence intervals (indicated by the shaded area) are larger for patients assessed as having lower LVEF than normal people (with higher LVEF).
(9)$$\mathrm{LVEF}\;=-1.614\times \frac{\mathrm{PEP}}{\mathrm{LVET}}+0.998$$
(10)LVEF =−1.169 Tei index× +1.043
(11)$$\mathrm{LVEF}\;=-1.476\times 0.5\times \left(\frac{\mathrm{PEP}}{\mathrm{LVET}}+\;\mathrm{Tei}\;\mathrm{index}\right)+1.070$$

### 3.2. ROC Classification

The LVEF assessment has clinical utility for cardiovascular syndrome classification and heart failure diagnosis [37]. According to the 2016 European Society of Cardiology (ESC) guidelines regarding the diagnosis and treatment of acute and chronic heart failure, the class of HFmrEF (heart failure with mid-range ejection fraction) was defined as an LVEF assessment larger than 40% but less than 50%. The other classes of HFrEF and HFpEF (heart failure with preserved ejection fraction) were retained [38].

The second proof-of-concept experiment for the DTW-based fiducial point detection approach was the application of the univariate ROC models to classify the subjects with HFrEF label listed in the disease column of Table 1. In Figure 10, four variables, including (1) LVEF, (2) PEP/LVET ratio, (3) Tei index, and (4) the mean PEP/LVET ratio and Tei index, were employed as the univariate predictors. To build the models, four data pairs, comprising diagnosis disease labels and the values of the predictors, were modeled by the GLMs as with the logit link function and binomial distribution function. The GLM-fitted probability vectors were set as scores and the “HFrEF” tag was set to as the positive label in the ROC curve configurations. Figure 10 exhibits the ROC classification result of four models to predict the diagnosis of HFrEF; extra indicators including the TP and FP rates, AUCs, and the optimal operating points of the predictors are also annotated. The optimal operating point was estimated at the condition that the classifier gave the best trade-off between the costs of failing to detect positives against the costs of raising false alarms.

The AUC of the LVEF model outperformed the other three models, with an AUC of 0.995 and the optimal operating point (LVEF cutoff at 0.4) at TP = 1 and FP = 0.029 (Figure 10a). This result conformed to the lower bound of HFmrEF of 40% suggested in the 2016 ESC guidelines. The AUC of the PEP/LVET ratio model was slightly greater than that of the Tei index model (with AUC = 0.937 > 0.928). In other words, the cardiologists had a 93.7% probability of correctly distinguish a patient with HFrEF from others with the aid of the PEP/LVET ratio model. Using the Tei index as the predictor, HFrEF could be expected to correctly diagnose 92.8% of the cases. As shown in Figure 10b,c, both models has the same TP and FP for the optimal operating points. Using the mean PEP/LVET ratio and Tei index as predictors improved the diagnostic prediction of any original predictors, with AUC = 0.949 (Figure 10d). In general classification evaluations, ROC model predictions with AUC values equal to 0.5 suggest no discriminative ability. AUC values between 0.7 and 0.8, between 0.8 and 0.9, and >0.9 indicate acceptable, excellent, and outstanding discriminative ability, respectively [39].

Through the proof-of-concept experiments of GLM models and ROC curve predictions, the CTIs and the cardiac indexes (PEP/LVET ration and Tei index) derived from the DTW-based quasi-synchronous alignment herein were confirmed to be reasonable in clinical practice. The optimal operating points corresponding to the last three ROC curves are also annotated in Figure 9. Moreover, the corresponding optimal decision points (0.4, 0.465, 0.436 and 0.45) for HFrEF assessment in the ROC curves were in Figure 10 for the comparison of the suggestion in 2016 ESC guidelines, LVEF = 0.4. Hopefully, these findings will provide reference for future study.

## 4. Discussion and Conclusions

The present study examined the issue of misalignment in the conventional quasi-synchronous alignment method and introduced the intermediary signals (reference signal pair of ECG and SCG) and DTW algorithm to eliminate the timing error in SCG fiducial point detection. The advantages and effects of the collaboration of reference signals and DTW algorithm as well as the new distance cost function were demonstrated in the graphical illustrations. The combination of the intermediary signals and DTW alignment in SCG fiducial point detection was proposed and realized for the first time.

It was known that SCG fiducial point delineation was hindered by artifact effects, the ambiguity in specific event waveforms and the lack of detection procedures. The proposed method abandoned the idea to look for the fiducial point co-occurring waveforms but seek for fusing heterogeneous modalities to allocate the fiducial points in the personal SCG reference signal. DTW algorithm was leveraged afterwards to project the fiducial points to non-reference signals. A merit of aligning signal pair with DTW algorithm is that it does not just align the prominent peaks or valleys but the entire signals. Because the extreme points serve as the anchor points during the alignment, points in between are enforced to be regulated. On the condition that the artifact does not override the signal waveform too much, DTW could overcome the distortion. Therefore, in case of minor signal distortion or featureless points are identified as the fiducial points, the projection can still function correctly with the assist of DTW.

This proposed DTW-based quasi-synchronous alignment is not only dedicated to SCG fiducial point detection but is also applicable to other scenarios. The non-ECG reference signal (SCG reference signal) and the application target (seismocardiography) could be changed to other cardiac signals, such as phonocardiography, ballistocardiography and impedance cardiography. The echocardiogram images could also be changed to other imaging modalities or other cardiac signal templates with the target cardiac events annotated. The application concept of the proposal is not limited in biomedical scenarios; it is possible to extend to the speech recognition or gesture identification and so on.

As the concept was proven clinically in the 56-individual dataset, the software program implemented based on the proposed flowchart achieved high prediction rates (>92.8%) in the experiment of distinguishing patients with HFrEF from others. With the verified diagnostic utility, the proposed quasi-synchronous fiducial point detection procedures could be further refined and standardized to expedite the development in SCG technology and shorten the path from bench to bedside.

Recently, machine learning has been one of the major topics in biomedical engineering; however, building a machine learning model for SCG fiducial point detection requires plenty of labeled training data. The proposed flowchart could be used as a framework and guidelines of the automatic program development for fiducial point detection. Therefore, the program-generated SCG fiducial point annotations could be used as the labeled training data set for the supervised machine learning process in alternative fiducial point detection approaches or other feature identification applications. In addition, by adding new routines, the framework can extend more features to the SCG research, such as signal template generation, signal morphology clustering and multichannel SCG applications. In the future, the framework may be further integrated into the cloud computing services together with the ambulatory ECG/SCG system for home-based real time health monitoring.

## Figures and Tables

**Figure 1 biosensors-12-00374-f001:**
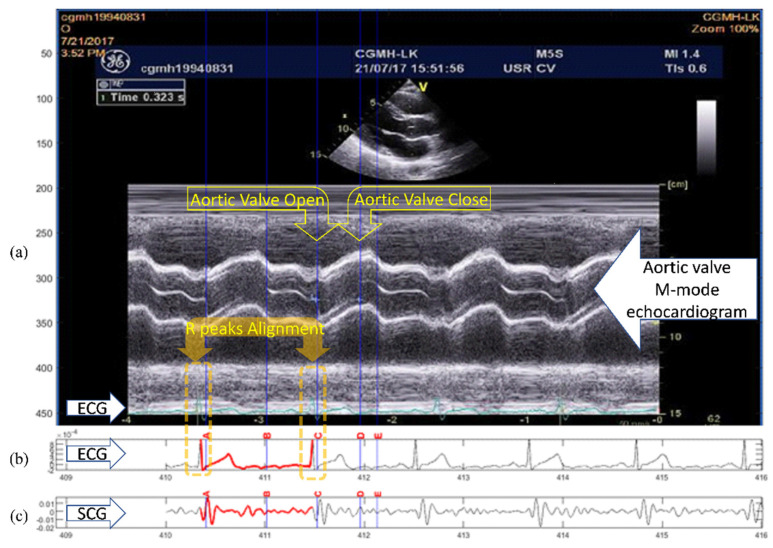
Example of the quasi-synchronous alignment for an echocardiogram image and an SCG measurement: (**a**) M-mode echocardiogram image of the aortic valve (with an ECG signal at the bottom of the image); (**b**) ECG signal simultaneously measured with SCG signal; (**c**) SCG signal.

**Figure 2 biosensors-12-00374-f002:**
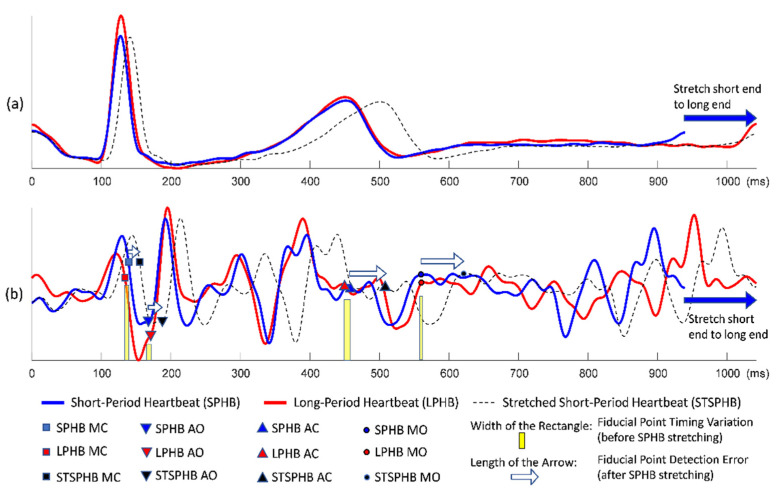
Conventional quasi-synchronous alignment by stretching the short-period signal: (**a**) Long-period (red), short-period (blue), and stretched short-period (black dashed) ECG signals; (**b**) Long-period (red), short-period (blue), and stretched short-period (black dashed) SCG signals with fiducial points and the detection error indicators (white arrows).

**Figure 3 biosensors-12-00374-f003:**
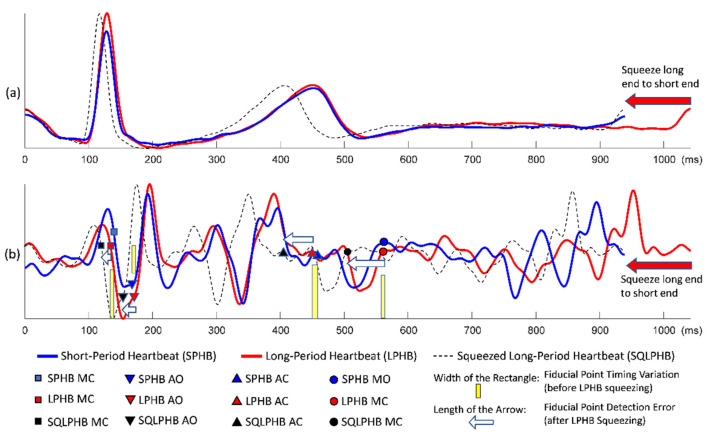
Conventional quasi-synchronous alignment by squeezing the long-period signal: (**a**) Long-period (red), short-period (blue), and squeezed long-period (black dashed) ECG signals; (**b**) Long-period (red), short-period (blue), and squeezed long-period (black dashed) SCG signals with fiducial points and the detection error indicators (white arrows).

**Figure 5 biosensors-12-00374-f005:**
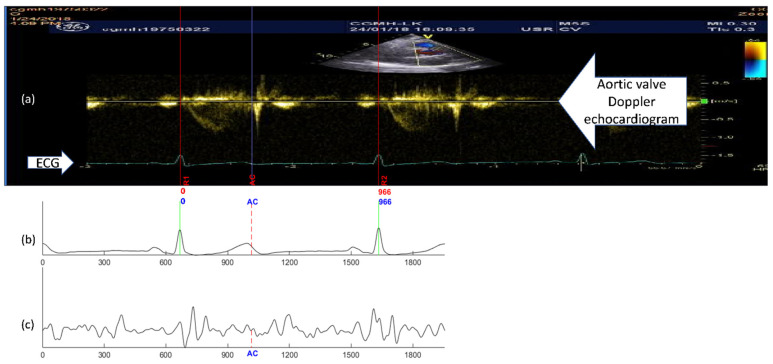
Illustration of aligning a reference signal pair to the selected echocardiogram section and projecting the specific cardiac event (aortic valve closing, AC) from echocardiogram to SCG curve as the detected fiducial point: (**a**) Echocardiogram image and the selected section enclosed within vertical lines marked with R1 and R2; (**b**) The aligned ECG of the reference signal pair; (**c**) The aligned SCG of the reference signal pair.

**Figure 6 biosensors-12-00374-f006:**
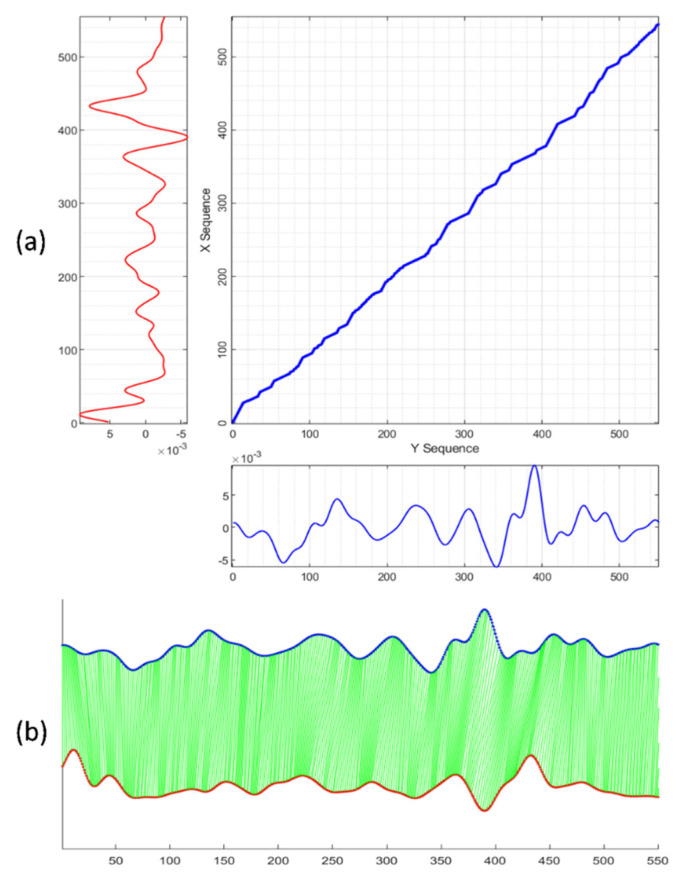
Illustration of the DTW alignment of two signal sequences: (**a**) Matrix grid view of DTW alignment; (**b**) Point-to-point mapping of DTW alignment.

**Figure 7 biosensors-12-00374-f007:**
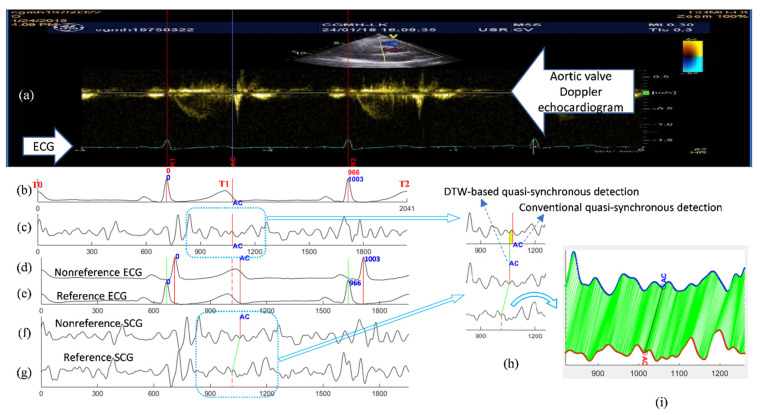
Illustrations and comparison of the fiducial point detection in the nonreference SCG signal between the methods of the conventional and the DTW-based quasi-synchronous alignment: (**a**) Doppler echocardiogram image with the identified ECG R peaks (R1, R2) and aortic valve closing (AC) event; (**b**) ECG alignment (to the ECG signal in the echocardiogram) under the conventional quasi-synchronization (aligned through shifting and rescaling); (**c**) SCG signal (synchronous to (**b**)) aligned by the conventional approach with the fiducial point AC detected by virtual line extending from the echocardiogram; (**d**) Nonreference ECG signal (no need for R peaks alignment to other ECG signal); (**e**) Reference ECG signal with R peaks aligned to ECG R peaks in the echocardiogram R1 and R2 (aligned by shifting only) under DTW-based quasi-synchronous alignment; (**f**) Nonreference SCG signal (synchronous to (**d**)) with the fiducial point AC projected by DTW-based quasi-synchronous alignment; (**g**) Reference SCG signal (synchronous to (**e**)) aligned to echocardiogram with the fiducial point AC detected by virtual line extending from the echocardiogram under DTW-based approach; (**h**) The detection error (indicated by the width of yellow rectangle in the upper graph) of the conventional method illustrated by comparing the AC points detected from conventional quasi-synchronous alignment (upper trace) and DTW-based (middle trace) approach; (**i**) Point-to-point mapping of the nonreference SCG signal (upper trace) with the reference SCG signal (lower trace) under DTW alignment.

**Figure 8 biosensors-12-00374-f008:**
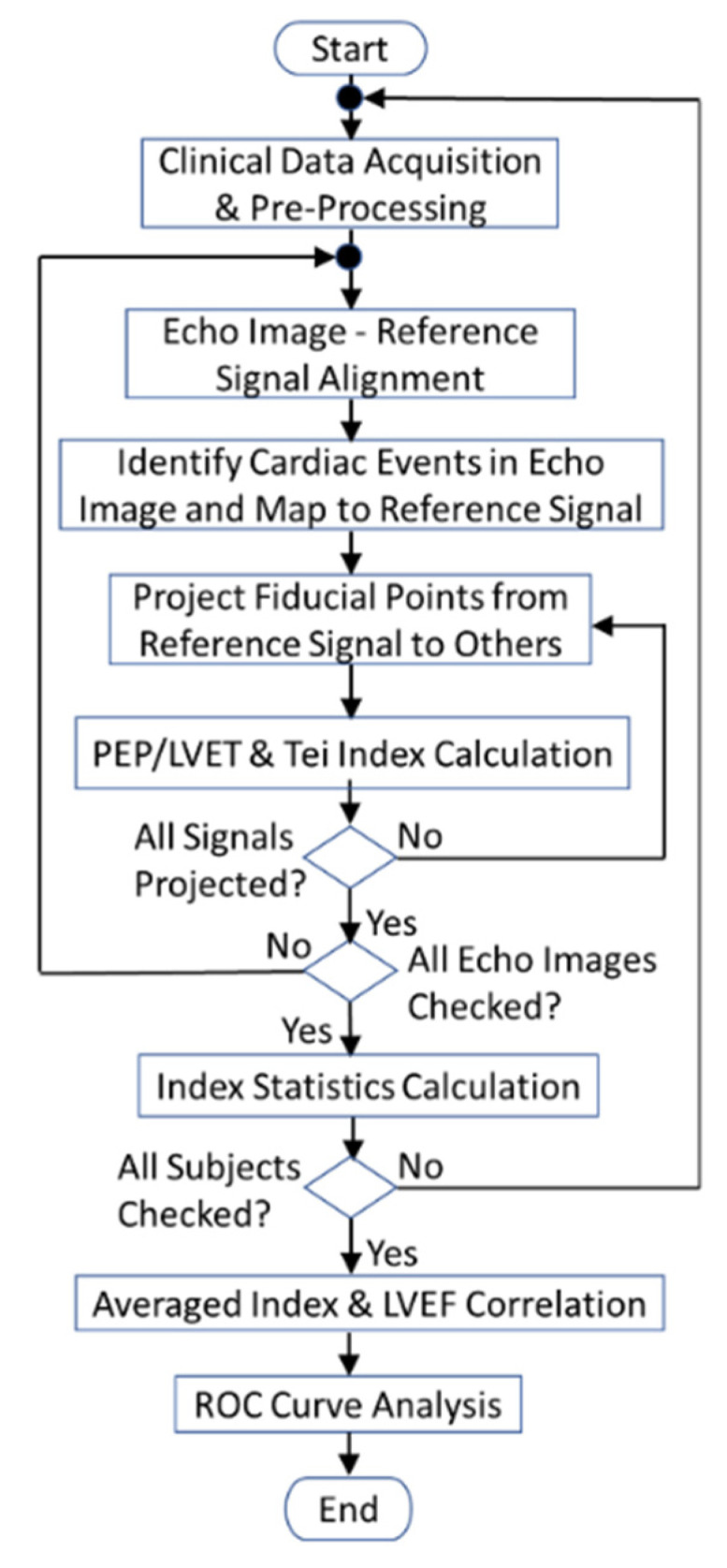
Programming flowchart of the DTW-based fiducial point detection and validation.

**Figure 9 biosensors-12-00374-f009:**
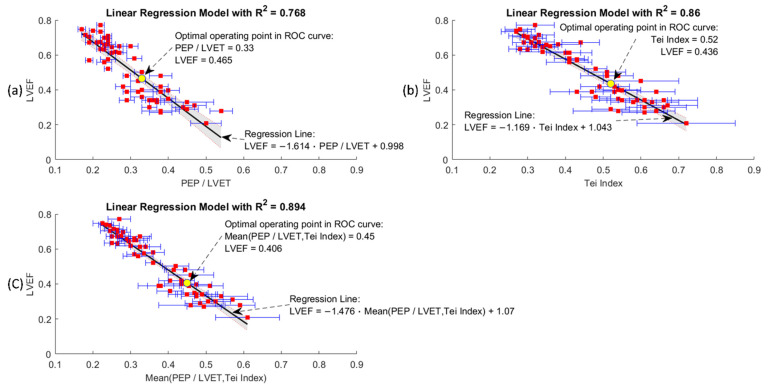
Linear regression models of clinical LVEF assessment versus: (**a**) PEP/LVET ration; (**b**) Tei index; (**c**) mean of PEP/LVET and Tei index. Each graph is annotated with mean (red squares) and standard deviation (blue error bars) of the variable from subjects, the 95% confidence interval of the regression line (shaded area), the coefficient of determination (R2) and the optimal operating point determined in the ROC curve analysis (yellow point).

**Figure 10 biosensors-12-00374-f010:**
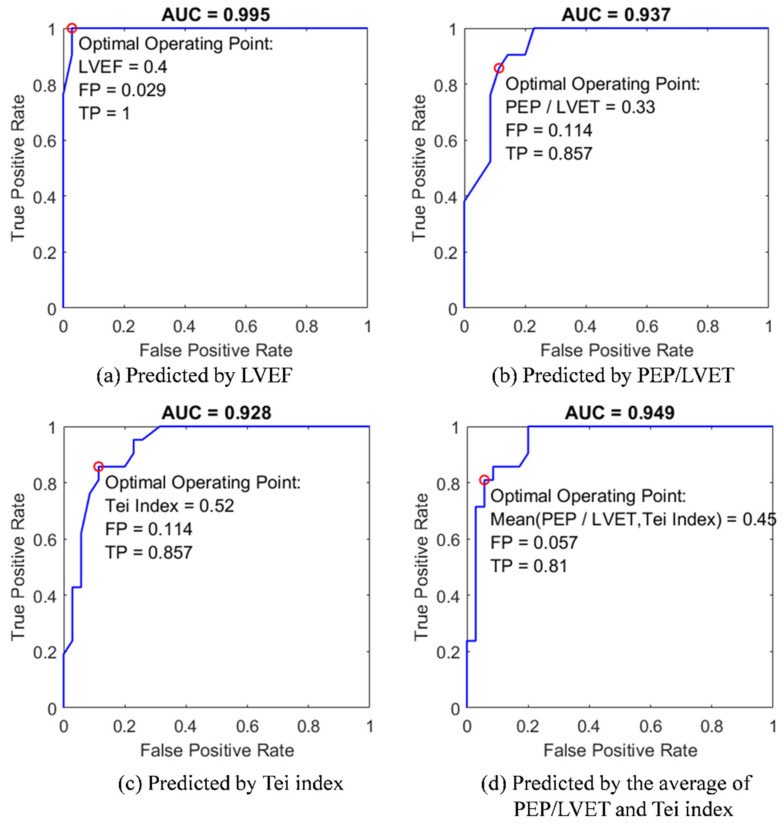
The comparison of ROC model analysis for the classification of patients with HFrEF by using four univariate predictors: (**a**) clinical LVEF assessment; (**b**) the PEP/LVET ratio; (**c**) Tei index; (**d**) the mean of PEP/LVET ration and Tei index.

**Table 1 biosensors-12-00374-t001:** Clinical assessment and SCG fiducial point derived cardiac parameters for GLM and ROC curve analysis.

Subject ID	Sex	Age	Clinical Assess			SCG Fiducial Point Derivative							Subject ID	Sex	Age	Clinical Assess			SCG Fiducial Point Derivative							Subject ID	Sex	Age	Clinical Assess			SCG Fiducial Point Derivative					
			LVEF	Disease		Cycles	PEP/LVET			Tei Index						LVEF	Disease		Cycles	PEP/LVET			Tei Index						LVEF	Disease		Cycles	PEP/LVET			Tei Index	
							Avg	SD		Avg	SD									Avg	SD		Avg	SD									Avg	SD		Avg	SD
Subject-01	F	22	29.0%	HFrEF		72	0.45	0.03		0.52	0.05		Subject-20	F	80	33.0%	HFrEF		39	0.37	0.02		0.58	0.06		Subject-39	M	46	67.2%	Normal		73	0.19	0.01		0.31	0.02
Subject-02	F	22	70.9%	Normal		85	0.24	0.01		0.29	0.04		Subject-21	F	82	45.0%	MI		62	0.32	0.02		0.60	0.10		Subject-40	M	48	67.3%	MI		75	0.21	0.01		0.44	0.05
Subject-03	F	23	77.1%	Normal		56	0.22	0.01		0.32	0.05		Subject-22	F	84	57.0%	MI		61	0.19	0.01		0.43	0.03		Subject-41	M	49	50.3%	HFpEF		70	0.33	0.02		0.51	0.05
Subject-04	F	24	63.6%	Normal		95	0.22	0.01		0.28	0.03		Subject-23	F	85	42.0%	HFmrEF		88	0.38	0.02		0.49	0.05		Subject-42	M	51	67.2%	Normal		78	0.24	0.01		0.29	0.03
Subject-05	F	26	71.5%	Normal		79	0.18	0.01		0.33	0.04		Subject-24	F	85	27.0%	HFrEF		89	0.38	0.03		0.61	0.06		Subject-43	M	48	55.8%	MI		66	0.23	0.02		0.41	0.06
Subject-06	F	26	74.0%	Normal		78	0.20	0.01		0.27	0.03		Subject-25	F	90	35.0%	HFrEF		60	0.40	0.02		0.54	0.05		Subject-44	M	58	34.0%	HFrEF		79	0.29	0.02		0.66	0.09
Subject-07	F	27	67.5%	Normal		83	0.22	0.01		0.33	0.04		Subject-26	F	91	39.8%	HFrEF		89	0.40	0.03		0.55	0.05		Subject-45	M	62	21.0%	HFrEF		66	0.5	0.04		0.72	0.13
Subject-08	F	34	34.0%	HFrEF		52	0.37	0.04		0.61	0.07		Subject-27	M	21	70.4%	Normal		62	0.19	0.01		0.30	0.04		Subject-46	M	62	33.0%	HFrEF		38	0.45	0.07		0.63	0.07
Subject-09	F	37	61.4%	Normal		81	0.27	0.02		0.41	0.06		Subject-28	M	23	65.9%	Normal		61	0.23	0.01		0.35	0.04		Subject-47	M	63	34.0%	HFrEF		80	0.36	0.02		0.59	0.08
Subject-10	F	43	61.3%	Normal		34	0.25	0.01		0.40	0.04		Subject-29	M	23	69.5%	Normal		90	0.24	0.01		0.32	0.03		Subject-48	M	63	31.0%	HFrEF		46	0.47	0.03		0.67	0.08
Subject-11	F	43	65.0%	Normal		98	0.27	0.02		0.36	0.02		Subject-30	M	24	66.2%	Normal		76	0.24	0.02		0.38	0.06		Subject-49	M	65	57.6%	MI		73	0.24	0.02		0.43	0.03
Subject-12	F	47	73.3%	Normal		64	0.22	0.01		0.27	0.04		Subject-31	M	24	69.9%	Normal		70	0.23	0.01		0.30	0.02		Subject-50	M	66	52.0%	MI		50	0.24	0.01		0.48	0.03
Subject-13	F	60	34.0%	HFrEF		75	0.35	0.02		0.55	0.08		Subject-32	M	24	64.4%	Normal		82	0.25	0.02		0.34	0.04		Subject-51	M	68	39.0%	HFrEF		54	0.32	0.04		0.43	0.07
Subject-14	F	60	42.0%	HFmrEF		65	0.28	0.02		0.53	0.04		Subject-33	M	25	62.0%	Normal		69	0.26	0.02		0.34	0.06		Subject-52	M	69	30.0%	HFrEF		75	0.44	0.03		0.61	0.05
Subject-15	F	68	28.0%	HFrEF		67	0.38	0.05		0.54	0.12		Subject-34	M	27	48.0%	HFmrEF		37	0.38	0.03		0.51	0.07		Subject-53	M	70	40.0%	HFrEF		40	0.33	0.03		0.54	0.06
Subject-16	F	68	65.0%	MI *		80	0.30	0.02		0.32	0.05		Subject-35	M	28	63.1%	Normal		62	0.23	0.01		0.30	0.03		Subject-54	M	79	30.0%	HFrEF		67	0.35	0.02		0.66	0.06
Subject-17	F	70	39.0%	MI		60	0.38	0.02		0.64	0.05		Subject-36	M	38	74.7%	MI		76	0.17	0.01		0.28	0.04		Subject-55	M	79	48.0%	MI		78	0.29	0.01		0.54	0.04
Subject-18	F	70	36.0%	HFrEF		38	0.33	0.03		0.48	0.04		Subject-37	M	44	58.0%	MI		74	0.31	0.02		0.41	0.04		Subject-56	M	88	39.0%	HFrEF		106	0.29	0.01		0.47	0.06
Subject-19	F	72	28.0%	HFrEF		84	0.54	0.03		0.64	0.05		Subject-38	M	45	39.0%	HFrEF		69	0.38	0.01		0.53	0.03													

* MI: Myocardial Infarction.

## Data Availability

Not applicable.

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
