# Peer review of "Computer-Aided Detection of Fiducial Points in Seismocardiography through Dynamic Time Warping"

_biosensors, 2022, doi:10.3390/bios12060374_

Round 1
Reviewer 1 Report
Analysis of wearable solutions for health monitoring is a challenging and promising topic. seismocardiography fiducial point detection can be performed with the aid of quasi-synchronous alignment between echocardiography images and seismocardiogram signals. However, signal misalignments have been observed. This study introduces reference signals and dynamic time warping algorithm to obtain the PEP/LVET and the Tei index. In their finding, favorable heterogeneity between these parameters and LVEF was obtained. In general, the study is well investigated. However, the authors may consider the following comments to further improve the manuscript:
- Is there any difference in the efficacy obtained by this algorithm between patients with severe cardiac arrhythmia or heart failure and normal people?
- It was known that SCG was limited by the ambiguity in specific event waveforms and the lack of detection procedures. Principle for DTW overcoming this problem requires more in-depth introduction or discussion (for instance, the specific characteristics of SCG to be recognized or ignored).
- Chest wall motion-based monitoring should be inevitably interfered by the body movements, friction, and even respiratory movements. How to eliminate the interference of these influencing factors?
- The real noninvasive wearable devices for long-term, home-based cardiac monitoring should be independent from other methods, including ECG and echocardiography. It needs to be discussed how to further improve the current algorithm to achieve this final goal.
Author Response
- Is there any difference in the efficacy obtained by this algorithm between patients with severe cardiac arrhythmia or heart failure and normal people?
- As shown in the revised Figure 9, the error bars indicate the standard deviations of PEP/LVET, Tei index and the mean of the above two. The standard deviations obtained from the algorithm are less for normal people (with larger LVEF values) while the standard deviations increase as the LVEF value decrease for people with severer cardiovascular syndromes. That means in case of fewer SCG measurements the diagnostic prediction could be still accurate for normal people, whereas for heart disease patients to obtain accurate assessment needs the statistical results from more measurements.
- New content has been added to the text. “The negative proportionalities in the graphs of Figure 9 are obvious, whereas the standard deviation (indicated by the blue error bars) and 95% confidence intervals (indicated by the shaded area) are larger for patients assessed as with less LVEF than those of the normal people (with higher LVEF).”
- It was known that SCG was limited by the ambiguity in specific event waveforms and the lack of detection procedures. Principle for DTW overcoming this problem requires more in-depth introduction or discussion (for instance, the specific characteristics of SCG to be recognized or ignored).
- Due to the morphology varies among people, not any waveform peaks or valleys were assumed as the candidates of the SCG fiducial points in the proposed method. This method does not identify fiducial points in “the reference SCG signal” by the specific morphological characteristics but relies on the visual identifications in the aligned echocardiogram image and then mapping the points to reference SCG signal.
- The content below has been added to the text. “It was known that SCG fiducial point delineation was hindered by the ambiguity in specific event waveforms and the lack of detection procedures. The proposed method abandoned the idea to look for the fiducial point co-occurring waveforms but seek for fusing heterogeneous modalities to allocate the fiducial points in the personal SCG reference signal. DTW algorithm was leveraged afterwards to project the fiducial points to non-reference signals. One merit of aligning signal pair with DTW algorithm is that it does not just align the prominent peaks or valleys but the entire signals. In case of featureless points are identified as the fiducial points, the projection can still function correctly by DTW.”
- Chest wall motion-based monitoring should be inevitably interfered by the body movements, friction, and even respiratory movements. How to eliminate the interference of these influencing factors?
- As a kind of the bio-signals, the SCG signals are non-stationary, therefore, to remove all the artifacts and respiratory effects from the SCG signals is not possible, though the signal processing procedures (detrend, wavelet denoising, and 1~40 Hz band pass filtering) have been applied in advance. On the other hand, this is why the DTW algorithm was introduced and dedicated to aligning signals according to their morphological resemblances attributed to different cardiac events.
- The real noninvasive wearable devices for long-term, home-based cardiac monitoring should be independent from other methods, including ECG and echocardiography. It needs to be discussed how to further improve the current algorithm to achieve this final goal.
- The real SCG alone home-based cardiac monitoring requires further investigations. Currently, heterogeneous modality cooperation is more feasible; that is diagnostic assessing by ECG and echocardiogram (help SCG fiducial point identification) while monitoring by ECG and SCG. Further developing milestones include 1) the intra- and inter-subject SCG signal template generations, 2) fiducial point annotation standardization, 3) the relationship between SCG signal morphologies and detection locations. The quasi-synchronous alignment method could be used as the core block, by adding different function blocks to achieve the milestones. Among the milestones, the intra-subject template generation is almost finished in our investigation.
- The contents below have been added to the text of introduction. “Heterogeneous modality cooperation serves as the alternative; that is to conduct the diagnostic assessment by ECG and echocardiogram (also assisting SCG fiducial point identification) while home monitoring by ECG and SCG.” & “Nevertheless, further milestones under development including 1) the intra- and in-ter-subject SCG signal template generations, 2) fiducial point annotation standardiza-tion, 3) the analysis of relationship between SCG signal morphologies and detection locations.”

Reviewer 2 Report
This paper introduced a novel signal processing framework for the fiducial points from SCG signals and presented two representative applications on data from HF patients with different levels of EF metrics. The main concept is to apply DTW to standardize waveforms and extract systolic time intervals that are supposed to be related to LVEF and other labels. Overall, the paper is well organized and the topic is of interest to the are of SCG analysis.
I have a couple of major and minor questions as follows:
1)The authors are encouraged to elaborate on their method differences compared to the work presented in [1] and [2] regarding their application of DTW to SCG signals. Note although they mentioned BCG in their paper, but they actually processed sternal vibrations, i.e., SCG.
[1]Javaid, Abdul Q., Hazar Ashouri, and Omer T. Inan. "Estimating systolic time intervals during walking using wearable ballistocardiography." In 2016 IEEE-EMBS International Conference on Biomedical and Health Informatics (BHI), pp. 549-552. IEEE, 2016.
[2] Javaid, A.Q., Ashouri, H., Dorier, A., Etemadi, M., Heller, J.A., Roy, S. and Inan, O.T., 2016. Quantifying and reducing motion artifacts in wearable seismocardiogram measurements during walking to assess left ventricular health. IEEE Transactions on Biomedical Engineering, 64(6), pp.1277-1286.
2) What is the sampling rate of the echo? One challenge of corresponding SCG waveforms with echo is that the sampling rate of echo is usually too low compared to SCG, which may result in timing inaccuracy due to timing resolution. Please comment.
3) Have the authors tried a multivariable model that fuses the presented features? Maybe a simple SVM or regression could see the data better.
4) The reviewer is concerned about subject bias in the current evaluation. Is there any leave out test?
Author Response
Comments and Suggestions for Authors
This paper introduced a novel signal processing framework for the fiducial points from SCG signals and presented two representative applications on data from HF patients with different levels of EF metrics. The main concept is to apply DTW to standardize waveforms and extract systolic time intervals that are supposed to be related to LVEF and other labels. Overall, the paper is well organized and the topic is of interest to the are of SCG analysis.
I have a couple of major and minor questions as follows:
1)The authors are encouraged to elaborate on their method differences compared to the work presented in [1] and [2] regarding their application of DTW to SCG signals. Note although they mentioned BCG in their paper, but they actually processed sternal vibrations, i.e., SCG.
[1]Javaid, Abdul Q., Hazar Ashouri, and Omer T. Inan. "Estimating systolic time intervals during walking using wearable ballistocardiography." In 2016 IEEE-EMBS International Conference on Biomedical and Health Informatics (BHI), pp. 549-552. IEEE, 2016.
[2] Javaid, A.Q., Ashouri, H., Dorier, A., Etemadi, M., Heller, J.A., Roy, S. and Inan, O.T., 2016. Quantifying and reducing motion artifacts in wearable seismocardiogram measurements during walking to assess left ventricular health. IEEE Transactions on Biomedical Engineering, 64(6), pp.1277-1286.
- It is an interesting task to assess the left ventricular health during walking. We would expect to challenge the ambulatory cases after the annotated SCG template generation study.
2) What is the sampling rate of the echo? One challenge of corresponding SCG waveforms with echo is that the sampling rate of echo is usually too low compared to SCG, which may result in timing inaccuracy due to timing resolution. Please comment.
- Since the proposed method adopted mostly the M mode echocardiogram to align with SCG reference signal (sampling rate 1KHz) in the same chart (computer screen), the temporary resolution is to be concerned. Per the description (refer to the bullets below) in the web link, the M mode echocardiogram is applicable to the multimodal (echo-ECG-SCG) cardiac event analysis.
- “Because M-mode images are updated 1000 times per second, they provide greater temporal resolution than two-dimensional (2D) echocardiography; thus, more subtle changes in motion or dimension can be appreciated.”
- https://www.sciencedirect.com/topics/nursing-and-health-professions/m-mode-echocardiography
3) Have the authors tried a multivariable model that fuses the presented features? Maybe a simple SVM or regression could see the data better.
- Not yet to investigate the HFrEF prediction with non-LVEF variable or multivariate ROC analysis. The main purpose of the univariate (derived from the detected SCG fiducial points) ROC classification is to reference to the current gold standard of the HFrEF diagnosis using LVEF (2016 ESC guidelines). However, applying multivariate models in the correlation analysis (LVEF versus multiple CTIs) or ROC classification may result interesting results for HF diagnosis. We could study this topic some other time.
4) The reviewer is concerned about subject bias in the current evaluation. Is there any leave out test?
- The leave out test was considered but conducted elsewhere. As the study is part of the achievements in our project of SCG automatic fiducial point annotation and template generation (refer to the function block diagram in the upload file), the leave out tests were planned in the white blocks in middle and the right-hand side.
- Currently we have observed little variation of PEP/LVET and the Tei index between the Train set and Test set. The results will be disclosed in our next paper.

Reviewer 3 Report
This research is very interesting and adequate to this Journal. Usin AI is possible improve better the effectiveness of the DTW-based quasi-synchronous alignment in seismocardiography fiducial point detection, as in:
Ismail Elnaggar, Tero Hurnanen, Olli Lahdenoja, Antti Airola, Matti Kaisti, Tuija Vasankari, Jouni Pykäri, Mikko Savontaus, Tero Koivisto:
Detecting Aortic Stenosis Using Seismocardiography and Gryocardiography Combined with Convolutional Neural Networks. CinC 2021: 1-4
Author Response
Comments and Suggestions for Authors
This research is very interesting and adequate to this Journal. Usin AI is possible improve better the effectiveness of the DTW-based quasi-synchronous alignment in seismocardiography fiducial point detection, as in:
Ismail Elnaggar, Tero Hurnanen, Olli Lahdenoja, Antti Airola, Matti Kaisti, Tuija Vasankari, Jouni Pykäri, Mikko Savontaus, Tero Koivisto:
Detecting Aortic Stenosis Using Seismocardiography and Gryocardiography Combined with Convolutional Neural Networks. CinC 2021: 1-4
- The CNN application in the article mentioned above is a novel approach to deal with the ECG, SCG and GCG signals in time and frequency domains.
- The original intention to adopt DTW is to reserve SCG signal morphology in the averaging process as the variations were seen not only in the amplitude but also in the temporal position. Machine learning could also serve for the same purpose. We have tried to use 1D-CNN to identify the SCG fiducial points but failed. Fortunately, the bonus of the present automatic fiducial point annotation tool is that a lot of labeled training data could be generated instead of manual generation. With enough training data, it may encourage us to investigate the machine learning application in cardiovascular disease diagnosis and prognosis.

Reviewer 4 Report
This submission manuscript is well-written. The authors designed the quasi-synchronous alignment method which combined the reference seismocardiography (SCG) signal and dynamic time warping method to detect the fiducial points. The results of fiducial point detection were validated by the contractility coefficient (PEP/LVET) and myocardial performance index (Tei index).
1) Is there any requirement of the SCG signal as a reference input for quasi-synchronous alignment? What type of artifact in the SCG signal would affect the detection accuracy? Please discuss it.
2) Are the linear correlation models validated? It could be better to provide the 95% confidence interval of the linear regression equations.
3) The abbreviation of "MI" in Table 1 should be explained.
4) In the result analysis section, it is recommended to discuss the differences of SCG fiducial point detection between MI and heart failure with reduced ejection fraction (HFrEF) diseases.
5) The criterion of optimal operating point should be interpret in the text.
Author Response
Comments and Suggestions for Authors
This submission manuscript is well-written. The authors designed the quasi-synchronous alignment method which combined the reference seismocardiography (SCG) signal and dynamic time warping method to detect the fiducial points. The results of fiducial point detection were validated by the contractility coefficient (PEP/LVET) and myocardial performance index (Tei index).
1) Is there any requirement of the SCG signal as a reference input for quasi-synchronous alignment? What type of artifact in the SCG signal would affect the detection accuracy? Please discuss it.
- Two type of quasi-synchronous alignment methods were discussed in the manuscript, the conventional and the DTW-based quasi-synchronous alignments. Only the latter one requires a reference signal pair (ECG and SCG). As far as the reference signal requirement is concerned, the heartbeat lengths of reference signals have to be as close to that of the echocardiogram image section as possible. Once the requirement was met, the DTW-based quasi-synchronous alignment could avoid stretching/squeezing in the alignment processing, therefore the signal distortion and misalignment were eliminated.
- In the conducted clinical trials, the subjects rested in the supine position during the SCG signal acquisition. Except for the vibrations come from the cardiac activities, the other signals were all artifacts, including the vibrations due to breathing, voluntary and involuntary body movements, equipment noise and other environmental or man-made noises. Many signal processing techniques could be used to remove some of the in-band and out-of-band (frequency) noise but some may still exist. DTW algorithm was adopted to ignore the residual noise and the non-stationary bio-signal behavior but to catch the major morphological resemblance for the timing analysis in cardiac time interval assessment. Statistical indicators were used instead of single heartbeat analysis to smooth out the impact of random artifacts.
- New content below has been added to the text. “It was known that SCG fiducial point delineation was hindered by the impacts from artifacts, the ambiguity in specific event waveforms and the lack of detection procedures. The proposed method abandoned the idea to look for the fiducial point co-occurring waveforms but seek for fusing heterogeneous modalities to allocate the fiducial points in the personal SCG reference signal. DTW algorithm was leveraged afterwards to project the fiducial points to non-reference signals. A merit of aligning signal pair with DTW algorithm is that it does not just align the prominent peaks or valleys but the entire signals. Because the extreme points serve as the anchor points during the alignment, points in between are enforced to be regulated. On the condition that the artifact does not override the signal waveform too much, DTW could overcome the distortion. Therefore, in case of minor signal distortion or featureless points are identified as the fiducial points, the projection can still function correctly assisted by DTW.”
2) Are the linear correlation models validated? It could be better to provide the 95% confidence interval of the linear regression equations.
- Yes, the linear correlation models were built by the generalized linear model library in Matlab 2020 (The MathWorks Inc.). The 95% confidence interval information was provided in the revised Figure 9.
3) The abbreviation of "MI" in Table 1 should be explained.
- Yes, the explanation (MI: Myocardial Infarction) has been added to the footer of Table 1.
4) In the result analysis section, it is recommended to discuss the differences of SCG fiducial point detection between MI and heart failure with reduced ejection fraction (HFrEF) diseases.
- In the conducted clinical trial, the disease column in Table 1 was based on the medical records and case history assessed by cardiologists. The HFrEF and MI were treated as different cardiac diseases (the co-exist cases were labeled as HFrEF). Whereas the experiments were designed to validate the concepts of the proposed fiducial point detection method, utilizing the guidelines in ESC 2016 for HFrEF diagnosis as the benchmarking, no similar guidelines are available for MI. Therefore, the different detection results between HFrEF and MI were not discussed. Further investigation is needed to address the feasibility of using the proposed method to differentiate HFrEF and MI.
5) The criterion of optimal operating point should be interpret in the text.
- Thanks for the notification. The interpretation below has been added to the text. “The optimal operating point was estimated at the condition that the classifier gave the best trade-off between the costs of failing to detect positives against the costs of raising false alarms.”

Round 2
Reviewer 1 Report
No further comments
